# Hemoglobin, albumin, lymphocyte and platelet (HALP) score for predicting early and late mortality in elderly patients with proximal femur fractures

**Abdussamed Vural**[1]*, **Turgut Dolanbay**[1], **Hilal Yagar**[2]

**1** Department of Emergency Medicine, Faculty of Medicine, Nigde Omer Halisdemir University, Nigde, Turkey, **2** Department of Orthopedics, Faculty of Medicine, Nigde Omer Halisdemir University, Nigde, Turkey

* abdussamedvural@gmail.com

## Abstract

**Data Availability Statement:** All relevant data are within the manuscript and its Supporting information files.

### Background

Predicting mortality and morbidity poses a significant challenge to physicians, leading to the development of various scoring systems. Among these, the hemoglobin, albumin, lymphocyte and platelet (HALP) score evaluates a patient's nutritional and immune status. The primary aim of this study was to determine the predictive effect of the HALP score on 30-day and 1-year mortality in elderly patients with proximal femoral fractures (PFFs).

### Materials and methods

Patient demographic, clinical, laboratory, and prognostic data were obtained. The patients were categorized into two groups: survival and nonsurvival at mortality endpoints. The HALP score was calculated and compared among the groups and with other mortality biomarkers such as C-reactive protein (CRP) and C-reactive protein to albumin ratio (CAR). The ability of the HALP score to predict mortality was compared between the groups. The mortality risk was also calculated at the optimal threshold.

### Results

The HALP score had a statistically significant predictive effect on mortality endpoints and was lower in the non-surviving group. The ability of the HALP score to predict 1-year mortality at the optimal cut-off value (17.975) was superb, with a sensitivity of 0.66 and specificity of 0.86 (AUC: 0.826, 95% CI: 0.784–0.868). In addition, the power of the HALP score to differentiate survivors and non-survivors was more significant than that of other indices (p < 0.001). Patients with a HALP score ≤ 17.975 had a 1-year mortality risk 11.794 times that of patients with a HALP score ≥ 17.975 (Odds ratio: 11.794, 95% CI [7.194–19.338], p < 0.001).

**Funding:** The author(s) received no specific funding for this work.

**Competing interests:** The authors have declared that no competing interests exist.

## Conclusion

The results indicate that the HALP score demonstrates efficacy and utility in predicting 30-day and 1-year mortality risk among elderly patients with PFFs.

## 1. Introduction

Proximal femoral fractures (PFFs) occur around the hip joint in the upper femur bone. The femoral head, neck, and intertrochanteric/subtrochanteric fractures are the main subcategories based on their location [1]. PFFs, one of the most common presentations to emergency departments (ED), usually occur because of falls and are more likely to be seen in the elderly, especially in patients with osteoporosis [2]. Mortality and morbidity prediction is a difficult issue for physicians, and various scoring systems have been developed. One of these scoring systems is the hemoglobin, albumin, lymphocyte and, platelet (HALP) score, which is a composite score used to assess the nutritional and immune status of a patient. It is considered a prognostic indicator and has been used to predict overall survival and other outcomes in various cancers and non-cancer diseases [3–7]. The HALP score, which is expressed as hemoglobin (g/L) × albumin (g/L) × lymphocyte (/L) / platelet (/L), was first used to predict the prognosis of gastric carcinoma by Chen et al. in 2015. In this study, low HALP score was reported as an indicator of poor prognosis [7]. In this respect, HALP scoring as a composite immunonutritional biomarker shows promise in assessing a patient's overall health status by integrating several routinely collected laboratory indicators.

C-reactive protein (CRP) and the C-reactive protein to albumin ratio (CAR) have emerged as significant inflammation-based biomarkers for assessing mortality risk in elderly patients with hip fractures. CRP is a well-established marker of systemic inflammation and has been shown to correlate with adverse outcomes in various medical conditions including hip fractures. Elevated preoperative CRP levels have been identified as an independent predictor of 1-year mortality in elderly patients undergoing hip fracture surgery [8, 9]. In addition to CRP, CAR has gained attention as a potential prognostic indicator. CAR provides insights into both the inflammatory and nutritional status of patients, making it a valuable prognostic factor for 30-day mortality following hip fractures [10]. In this study, we aimed to perform a comprehensive analysis comparing the ability of CRP, CAR and HALP scores to predict mortality outcomes. This is because each biomarker provides different clinical information and these inflammation-based markers have the potential to offer a more comprehensive approach to mortality risk, especially in older patients. The HALP score, which includes hemoglobin, albumin, lymphocyte count and platelet count, provides information on inflammation and nutritional status. However, as CRP and CAR can directly reflect levels of systemic inflammation, their comparison with the HALP score may help to develop more effective management strategies for elderly patients with hip fractures. Furthermore, CRP and CAR, like the HALP score, are practical parameters that can be easily used to predict mortality in various diseases.

Commonly used scoring systems for mortality due to hip fractures include the Physiologic and Operative Severity Score for Mortality and Morbidity Measurement (POSSUM) and Portsmouth POSSUM (P-POSSUM), Nottingham Hip Fracture Score (NHFS), Sernbo score, Older Person's Emergency Risk Assessment (OPERA), Shizuoka Hip Fracture Prognostic Score (SHiPS) and Multidimensional Prognostic Index (MPI) [11–16]. These tools were developed to predict mortality rates at different time points, including 30-day, 1-year and long-term mortality. These tools are useful for risk stratification, decision making and improving patient

outcomes for hip fractures. However, some studies have emphasised the limitations of these risk scores. For example, Jonsson et al. reported that the POSSUM score and NHFS were poor predictors of 30-day mortality in patients with hip fracture; mortality and morbidity in this cohort were largely due to factors not included in these scores [17]. Ramanathan et al. also reported that there is a risk of misinterpretation of the POSSUM score in very old hip fracture patients due to physiological changes in this group, and that this scoring cannot accurately predict the postoperative complication rate and mortality in this cohort [18]. Besides, these scoring systems rely on extensive information about comorbidities, the nature of the surgical intervention and the clinical, demographic and physiologic characteristics of patients, which can be difficult to obtain in all elderly patients in acute settings. Instead of using various indicators and scores to measure the overall perioperative health status of patients, a more practical and convenient scoring system is needed. In this context, the HALP score based on hemoglobin, albumin, lymphocyte and platelet levels in frail elderly patients with hip fracture is a practical tool that can be quickly assessed even at emergency admission as a predictor of mortality in this cohort group. Since HALP is based on laboratory tests, it can be used more quickly and easily.

The main hypothesis of this study was that a low HALP score in elderly patients with PFFs can be interpreted as an unfavorable prognostic indicator. The primary aim of this study was to determine the predictive effect of the HALP score on 30-day and 1-year mortality in elderly patients with PFFs admitted to the ED. The secondary objective is to provide a comprehensive approach to inflammation-based mortality prediction by comparing the HALP score with CRP and CAR in predicting these mortality endpoints.

## 2. Materials and methods

### 2.1. Ethical issues

The study was initiated after the approval of the Non-Interventional Ethics Committee of Nigde Omer Halisdemir University, Faculty of Medicine (decision no. 2023/72 dated 12/10/2023). This study was conducted in accordance with the Declaration of Helsinki (2013 revision). The researchers did not have access to any information that could reveal the identity of individual participants during or after the data collection process.

### 2.2. Study design and patient selection

Our medical facility directs individuals presenting with hip fractures to the ED for an initial evaluation and treatment. Subsequently, the patients were referred to the orthopedic department for surgical intervention. To conduct our retrospective analysis, we used both our hospital's digital medical record and physical patient files. Patient demographic, clinical, and laboratory data were extracted from the Hospital Information Management System and recorded in a Microsoft Excel spreadsheet. The data set of the study is provided in S1 File. The data for the study were accessed on May 20, 21, and 22, 2024 for research purposes. We narrowed our focus first to individuals diagnosed with hip fractures then femoral neck fractures who had undergone surgical repair. Patients were categorized into two groups: survival and non-survival at 30-day and 1-year mortality edpoints. The power of the HALP score in predicting mortality was compared between the groups using appropriate statistical tests. The Death Notification System of the Ministry of Health was used to determine the 1-year mortality of the patients. In addition, the power of the HALP score in predicting 1-year all-cause mortality in patients was compared with the CRP and CAR. In addition, survival analysis of the HALP score and other indices at optimal thresholds was performed, and the biomarkers were compared with each other.

**2.2.1. Operational definitions.** The HALP score was calculated according to the parameters in the hemogram and biochemistry blood results obtained at the time of ED admission. Similar to the literature, the HALP score was calculated as follows: hemoglobin (g/L) × albumin (g/L) × lymphocytes (/L) / platelets (/L).

CRP level (mg/dl): C-reactive protein level at admission time in the ED

CAR: CRP/ albumin ratio at admission time in the ED

In the survival analysis, the cutoff points for HALP (17.975), CRP (28.145), and CAR (0.6574) were determined by maximizing the Youden index (sensitivity + specificity—1) from ROC analysis. ROC curves were employed to identify optimal thresholds for predicting early and late mortality by evaluating each biomarker's diagnostic performance in terms of sensitivity (true positive rate) and specificity (true negative rate). The Youden index was chosen as it provides the best balance between these two metrics, ensuring maximal discrimination between mortality and survival.

This approach ensures that the cutoff values for HALP, CRP, and CAR offer the greatest predictive accuracy. By using the Youden index, we optimized the thresholds to enhance sensitivity and specificity, making them highly suitable for clinical use, where reliable prognostication is critical. Thus, these specific thresholds were data-driven, selected based on their ability to provide the most accurate mortality predictions, as confirmed by Kaplan-Meier survival analysis and the Log-rank test.

**2.2.2. Inclusion and exclusion criteria for the study.** This study included patients over 65 years of age who underwent surgery for proximal femoral neck fractures between November 1, 2020, and May 1, 2024. The study excluded patients with missing data; those with non-proximal femoral neck fractures such as femoral head, shaft, or pelvic fractures; those who left the ED voluntarily or were referred to other facilities; patients with malignancies, and with those severe metabolic and infectious conditions (end-stage renal and hepatic failure, sepsis) that could affect the HALP score.

## 2.3 Statistical analysis

Descriptive statistics and continuous data were expressed as mean and standard deviation (SD) or median value and interquartile range (IQR, 25–75). Categorical variables were presented as frequency and percentage. Quantitative data were tested for normal distribution using the Kolmogorov–Smirnov test. It was found that only hemoglobin and age variables were normally distributed. Pearson $\chi2$, or Fisher's exact test was used for intergroup comparisons in the qualitative data of patients categorized into two groups: survival and non-survival. The independent sample Student's t-test was used for intergroup comparisons of continuous quantitative data with a normal distribution, and the Mann–Whitney U test was used for non-normally distributed data. The optimum threshold value of the HALP score and other indices for mortality prediction was determined by ROC analysis and Youden index maximization. In addition, the area under the receiver operating characteristic curve (AUC) was calculated, and the sensitivity and specificity of the optimal threshold values of the HALP score and other indices were calculated. In addition, survival analysis of patients was performed using the Kaplan–Meier method and the log-rank test according to the optimum threshold value of the HALP score and other indices. In addition, univariable logistic regression analysis was performed to calculate the 1-year mortality risk between the living and deceased groups. Because our study was planned retrospectively, all patients who met the inclusion and exclusion criteria were included in the study, and a priori power analysis was not performed in the sample size calculation. However, the study power was determined by a post hoc test (t test) performed after the study. In the statistical analysis, all data were analyzed at a 95% confidence level, and a *p*

value < 0.05 was considered statistically significant. All data were entered into an Excel database (Microsoft Office 2010, Redmond, WA, USA), and the Statistical Package for Social Science (SPSS) program (IBM SPSS Statistics Version 26, SPSS Inc., Armonk, NY, USA) was used for statistical analysis. In the post-hoc power analysis, the statistical power of comparisons between survivors and non-survivors based on HALP, CRP and CAR levels, the analysis yielded a power of 0.9977 (99.77%) using a two-tailed t-test for independent groups, with an effect size (Cohen's d) of 0.5 (medium effect), an alpha level of 0.05 and a sample size of 250 for the survivor group and 146 for the non-survivor group.

## 3. Results

A total of 435 patients who met the eligibility criteria were included in the analysis. The mean age of the patients was 81.56 ± 7.37 years. Of the patients, 297 (68.3%) were female and 138 (31.7%) were male. 257 patients (59%) had femoral neck fractures and 178 patients (41%) had intertrochanteric fractures. All patients underwent hemiarhtoplasty. Of the patients included in the study, 390 (89.7%) had at least one chronic disease. The most common comorbidities were hypertension (83%), diabetes mellitus (39%), chronic obstructive pulmonary disease (32%), coronary artery disease (28%) and stroke (22%). The median 1-year survival after surgery was 68 days (21–210.75) (ranged from 1–353) and the median hospital stay was 7 days (5–8) (ranged from 1–21). In-hospital, one-month, three-month, six-month, and 1-year mortality rates were 3.7% (n = 16), 11% (n = 48), 19.1% (n = 83), 25.1% (n = 109), and 33.6% (n = 146), respectively. A significant difference was found in the survival status at 1-year mortality endpoint according to age but not sex and length of hospital stay. Table 1 summarizes these clinical and laboratory characteristics of patients with PFFs.

Furthermore, the length of hospital stay did not differ according to the early and late mortality endpoints (1-month and 1-year mortality) ($p = 0.165$ and $p = 0.931$, respectively). The comparison of the HALP score, CRP, and CAR index with 30 day and 1-year mortality is given in Table 2. Accordingly, HALP score, CRP and CAR had a statistically significant predictive effect on the mortality endpoints but HALP score was more accurate for predicting 30-day mortality.

The optimum cut-off values and ROC analyses of the HALP score and other indices calculated by maximizing the Youden index (sensitivity+specificity-1) for 1-year long-term mortality prediction are summarized in Table 3. According to these results, the ability of the HALP score was superior than other inflammatory indices to predict 1-year mortality at the calculated optimal cut-off value (17.975) was excellent, with a sensitivity of 0.66 and specificity of 0.86 (AUC: 0.826, 95% CI: 0.784–0.868).

In addition, there was 100% mortality below a HALP = 10.77, and 0% mortality above HALP = 39.95. At a cut-off value of 17.975, according to survival analysis using the Kaplan–Meier method (Log-rank test), the power of the HALP score to differentiate survival and non-survival groups was more significant than other indices ($p < 0.001$). This is shown in Fig 1.

Finally, according to univariable binary logistic regression analysis, patients with a HALP score ≤ 17.975 had a 1-year mortality risk 11.794 times that of patients with a HALP score ≥ 17.975 (OR: 11.794, 95% CI [7.194–19.338], p < 0.001); the risk of patients with CRP > 28.145 was 3.714 times that of patients with CRP ≤ 28.145 (OR: 3.714, 95% CI [2.417–5.705], p < 0.001) and the risk of patients with CAR > 0.6574 was 4.139 times that of patients with CAR ≤ 0.6574 (OR: 4.139, 95% CI [2.685–6.380], p < 0.001) (Table 4). This showed that the HALP score was a stronger predictor of mortality compared to CRP and CAR.

**Table 1. Demographic, clinical and laboratory characteristics of the patients.**

| Variables | Total | Survival at | Non-survival | P-value |
|---|---|---|---|---|
| | n = 435 | 1-year, n = 250 | at 1-year, n = 146 | |
| Age (years), ± SD | 81.56 ±7.37 | 80.26 ±7.02 | 83.60 ±7.61 | < 0.001* |
| Sex, n (%) | | | | |
| Male | 138 (31.7) | 81 (65.9) | 42 (34.1) | 0.451** |
| Female | 297 (68.3) | 169 (61.9) | 104 (38.1) | |
| Comorbidity¶ | | | | |
| Yes, n (%) | 390 (89.7) | 223 (62.5) | 134 (37.5) | 0.511** |
| No, n (%) | 45 (10.3) | 27 (69.2) | 12 (30.8) | |
| Mortality, n (%) | | - | - | N/A |
| In-hospital | 16 (3.7) | | | |
| 30-day | 48 (11) | | | |
| 3-month | 83 (19.1) | | | |
| 6-month¶¶ | 109 (25.1) | | | |
| 1-year¶¶ | 146 (33.6) | | | |
| 1-year survival time (days) | - | - | | N/A |
| (median, IQR: 25–75) | | | 68 (21–210.75) | |
| Length of hospital stay (day) | 7 (5–8) | 7 (5–8) | 6 (4–9) | |
| (median, IQR: 25–75) | | | | 0.931*** |
| Fracture types, n (%) | | | | |
| Femoral neck | 257 (59.1) | 156 (62.4) | 83 (56.8) | 0.126** |
| Intertrochanteric | 178 (40.9) | 94 (37.6) | 63 (43.2) | |
| Applied treatment, n (%) | | | | |
| Hemiarthroplasty | 435 (100) | 250 (100) | 146 (100) | N/A |
| HALP score, median (IQR, 25–75) | 23.35 (16.22–33.65) | 20.70 (17.34–25.31) | 15.39 (10.57–22,52) | < 0.001*** |
| CRP, median (IQR, 25–75) | 13 (2.3–57.8) | 8.65 (1.80–36.82) | 40.05 (7.37–100.82) | < 0.001*** |
| CAR, median (IQR, 25–75) | 0.36 (0.56–1.7) | 0.21 (0.04–1.08) | 1.39 (0.21–3.19) | < 0.001*** |

Abbreviations: IQR, interquartile range; N/A, non-applicable; HALP, hemoglobin, albumin, lymphocyte, platelet; CRP, c-reactive protein; CAR, c-reactive-to-albumin ratio; SD, standard deviation

* t-test,

** chi-square (pearson or Continuity Correction),

*** Mann–Whitney U

¶ The most common chronic conditions observed within the study cohort were hypertension, diabetes mellitus, chronic obstructive pulmonary disease, coronary artery disease, and stroke. The cohort was further stratified based on the presence or absence of at least one of these comorbidities.

¶¶ There were censored data. Sixteen patients were excluded from the 6-month mortality calculation and 39 from the 1-year mortality calculation. These individuals were considered censored data as they had not reached six months and one year post-surgery for femoral neck fracture at the time of data analysis.

## 4. Discussion

Mortality biomarkers are of great interest in clinical research. In this respect, many biochemical markers have been the subject of scientific research in the literature, in addition to the clinical characteristics of patients and risk scoring to be used in the prediction of disease-specific or all-cause mortality [19]. Although the HALP score is noteworthy as a useful biomarker, especially in cancer patients with low immunity and nutrition, anemia, and thrombosis tendency, it may be appropriate to use it in appropriate modeling in non-cancer diseases, as in our study. The frequency of anemia, immunodeficiency, malnutrition and the risk of intravascular thrombosis increases with age [20–23]. Older adults experience anemia, which has been identified as a critical factor contributing to the increased hip fractures and mortality in this

**Table 2. Comparison of the HALP score and other indices by mortality status at early and late term endpoints.**

| Factors | Endpoints (mortality) | Mortality status | n | Mean Rank | Mean ± SD | P-value |
|---------|----------------------|------------------|-----|-----------|-----------|---------|
| HALP | 30-day | Yes | 48 | 147.34 | 18.5 | < 0.001 |
|  |  | No | 387 | 226.76 | 28.27 |  |
|  | 1-year | Yes | 146 | 117.97 | 16.94 | < 0.001 |
|  |  | No | 250 | 245.53 | 33.3 |  |
| CRP | 30-day | Yes | 48 | 270.96 | 63.64 | 0.002 |
|  |  | No | 387 | 211.43 | 36.42 |  |
|  | 1-year | Yes | 146 | 243.98 | 62.69 | < 0.001 |
|  |  | No | 250 | 171.94 | 28.93 |  |
| CAR | 30-day | Yes | 48 | 275.44 | 2.06 | 0.001 |
|  |  | No | 387 | 210.88 | 1.11 |  |
|  | 1-year | Yes | 146 | 247.84 | 2.04 | < 0.001 |
|  |  | No | 250 | 169.68 | 0.83 |  |

All the p- values were calculated using Mann–Whitney U test

Abbreviations: HALP, hemoglobin, albumin, lymphocyte, platelet; CRP, c-reactive protein; CAR, c-reactive protein to albumin ratio

population [24, 25]. The demanding rehabilitation required following such fractures is often compromised by anemia, resulting in decreased functional capacity and reduced quality of life, ultimately leading to higher mortality rates [26]. Hypoalbuminemia is associated with elevated postoperative mortality risk due to insufficient protein reserves for tissue repair and immunological defense against infections [27]. Low lymphocyte counts suggest a weakened immune response, increasing mortality risk as these patients are prone to infections, which can cause severe complications post-surgery [28]. In addition, thrombocytopenia raises the risk of bleeding and postoperative complications, while high platelet counts may signal inflammatory conditions, also increasing mortality risk [29]. Therefore, the HALP score seems to be very important in assessing the health status of patients with PFFs; a low HALP score indicates a high risk of mortality. In the present study, a composite formulation of the HALP score was used, and the main finding was that the HALP score was useful in predicting both 30-day and 1-year all-cause mortality in elderly patients with PFFs. In addition, when the area under the curve in long-term mortality was evaluated, the predictive ability of the AUC value of the HALP score in 1-year mortality was found to be superior to that of CRP and CAR. In a study by Zheng et al. [30] on coronary artery disease, the ability of the HALP score to predict all-cause mortality was found to be satisfactory (AUC: 0.610); in a study by Akbas et al. [5] on acute ileus patients, the ability to detect underlying malignant causes of ileus was found to be excellent (AUC: 0.86). In our study, the ability of the HALP score to predict 1-year mortality was found to be very good, with an AUC of 0.826, similar to other studies.

**Table 3. AUC, sensitivity, specificity, and cut-off point of HALP score, CRP, and CAR for 1-year mortality.**

| Mortality at 1-year | AUC | 95% CI | Sensitivity | Specificity | Youden's index | Cut-off point |
|---------------------|-----|--------|-------------|-------------|----------------|---------------|
| HALP | 0.826 | 0.784–0.868 | 0.660 | 0.863 | 0.523 | 17.975 |
| CRP | 0.682 | 0.627–0.737 | 0.610 | 0.704 | 0.314 | 28.145 |
| CAR | 0.697 | 0.643–0.752 | 0.644 | 0.696 | 0.340 | 0.6574 |

Abbreviations: AUC, area under curve; CI, confidence interval; HALP, hemoglobin-albumin-lymphocyte-platelet; CRP, c-reactive protein; CAR, c-reactive protein to albumin ratio

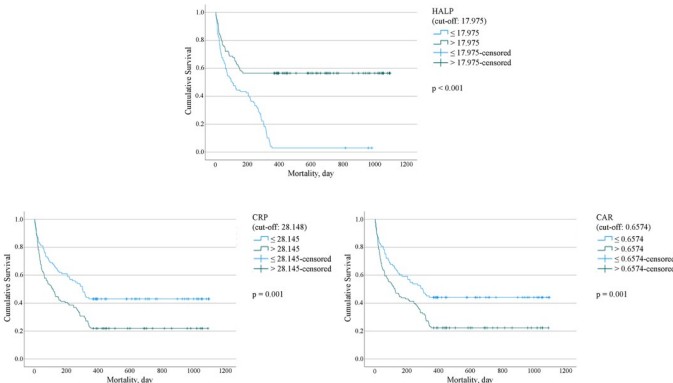

**Fig 1. Survival analysis (Kaplan-Meier method).** Log-rank survival function tests show the differences between HALP, CRP, and CAR levels to predict 1-year survival. For HALP, chi-square: 49.543, $p < 0.001$ at the cut-off point 17.975; for CRP, chi-square: 11.307, $p = 0.001$ at the 28.145; for CAR, chi-square:10.115, $p = 0.01$ at the cut-off 0.6574. Abbreviations: HALP, hemoglobin-albumin-lymphocyte-platelet; CRP, C-reactive protein; CAR, C-reactive protein to albumin ratio.

Farag et al. analyzed 32 tumor studies associated with HALP score. According to their results, the HALP score can be used as an independent risk factor to predict poor prognostic outcomes and tumor grading or differentiation [31]. The median HALP score among all tumor subtypes in this review was 31.2, with some cancer subtypes having higher (gastric cancer: 56.8, adenocarcinoma: 44.56) or lower (colorectal: 26.5, bladder: 22.2) median scores. Although there is no standardized and objective threshold that can be valid for all cancers, low HALP scores have been found to be statistically significant for poor outcome in intergroup comparisons in most cancers [7, 32–37]. In our study, the median HALP score was 23.35. This shows the heterogeneity of the HALP score. In addition, the HALP score was significantly lower in the non-survivor group at 30-day and 1-year mortality.

The relationship between HALP score and mortality has been studied in various medical contexts. Tian et al. [38] found a correlation between HALP scores and poor outcomes in acute ischemic stroke patients. Peng et al. [35] reported a strong correlation between HALP scores and cancer-specific survival in renal cell carcinoma patients, while Ruiz et al. [39] did not find a significant association between HALP scores and disease-free survival in colon cancer patients. Njoku et al. [40] also did not find a correlation between HALP scores and overall, cancer-specific, or recurrence-free survival in endometrial cancer patients. In addition, HALP has also shown remarkable prognostic ability in predicting survival and as a disease severity index in some non-cancer studies [41, 42]. In our study, the 1-year mortality risk of patients with a HALP score $< 17.975$ was approximately 12 times that of patients with a HALP score $> 17.975$.

**Table 4. Univariable logistic regression analysis of 1-year mortality risk based on HALP score, CRP and CAR.**

| Factors | Exp (B) (Odds ratio) | 95% C.I.for Exp (B) | P-value |
|---|---|---|---|
| | | Lower-Uppe r | |
| HALP ≤ 17.975 | 11.794 | 7.194–19.338 | < 0.001 |
| CRP > 28.145 | 3.714 | 2.417–5.705 | < 0.001 |
| CAR > 0.6574 | 4.139 | 2.685–6.380 | < 0.001 |

Abbreviations: HALP, hemoglobin-albumin-lymphocyte-platelet; CRP, C-reactive protein; CAR, C-reactive protein to albumin ratio; C.I, confidence interval

In a study by Balta et al. on CRP-based biomarkers (CRP, CAR, and C-reactive protein-to-lymphocyte ratio) as predictors of mortality in elderly patients with hip fractures, CRP was reported to be a good predictor of 30-day mortality after hip fracture surgery. Therefore, they recommended CRP and albumin control during routine preoperative anesthesia preparation [43]. Another study also showed the predictive value of CAR, particularly in the context of hip fractures, where high CAR values were associated with an increased risk of mortality [44]. In the current study, CRP and CAR had a statistically significant predictive effect on mortality endpoints, but the HALP score had a higher predictive power for 30-day and 1-year mortality. The 1-year survival analysis of the HALP score was statistically more significant than those of the other biomarkers. Therefore, the HALP score can be considered as a useful biomarker for differentiating between patients who die and those who do not die at the 1-year mortality point. This scoring system could potentially aid clinicians in making more informed decisions regarding treatment plans and resource allocation for elderly patients with PFFs. The HALP score's predictive capabilities for both short-term and long-term mortality risk underscore its versatility as a prognostic tool. Further validation studies across diverse patient populations and healthcare settings may help establish the HALP score as a standard assessment method in geriatric orthopedic care.

### 4.1. Limitations

This study has several important limitations. First, this is a retrospective observational study, which means that a direct causal relationship between HALP scores and mortality could not be established. Second, the potential confounding variables that may affect mortality outcomes in specific subgroups were not addressed. Differences in demographics (such as age, sex, and ethnicity), and surgical techniques might have significantly influenced the observed outcomes. Additionally, factors related to rehabilitation, medication use, nutritional status, and psychosocial aspects were not analyzed in detail. These unaccounted variables may limit the generalizability of our findings, which suggests that the observed associations should be interpreted with caution. Future studies should incorporate these factors to enhance our understanding of the relationship between HALP score and mortality risk in elderly patients with hip fractures. Moreover, more comprehensive and long-term prospective cohort studies are required to confirm the causal relationship between HALP levels and long-term mortality. Third, The fact that this study was conducted at a single center may also restrict the generalizability of the results. Additionally, focusing exclusively on elderly patients with PFFs may limit the applicability of the findings to other age groups or different types of fractures. Given these limitations, the prognostic value of the HALP score requires further validation in large-scale studies that include diverse populations.

## 5. Conclusion

The findings from the study show that the HALP score is effective in predicting 30-day and 1-year mortality risk in elderly patients with PFFs. Furthermore, the HALP score was found to be a stronger predictor of mortality than other biomarkers, such as CRP and CAR. In conclusion, this study shows that the use of the HALP score in the management and treatment of PFFs in elderly patients can play an important role in determining the prognosis of patients and devising appropriate treatment plans. The findings from our study may guide clinical practice and help improve patient health outcomes.

## Supporting information

**S1 File. Data set of the study.**
(XLSX)

## Author Contributions

**Conceptualization:** Abdussamed Vural, Turgut Dolanbay, Hilal Yagar.

**Data curation:** Abdussamed Vural, Turgut Dolanbay, Hilal Yagar.

**Formal analysis:** Abdussamed Vural, Turgut Dolanbay, Hilal Yagar.

**Investigation:** Abdussamed Vural, Hilal Yagar.

**Methodology:** Abdussamed Vural.

**Resources:** Abdussamed Vural, Hilal Yagar.

**Software:** Abdussamed Vural, Turgut Dolanbay.

**Supervision:** Abdussamed Vural, Hilal Yagar.

**Validation:** Turgut Dolanbay, Hilal Yagar.

**Visualization:** Abdussamed Vural, Turgut Dolanbay, Hilal Yagar.

**Writing – original draft:** Abdussamed Vural.

**Writing – review & editing:** Abdussamed Vural, Turgut Dolanbay, Hilal Yagar.

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
