## [Decision Letter · Decision Letter 0]

2 Oct 2024

PONE-D-24-38419HALP score for predicting early and late mortality in elderly patients with proximal femur fracturesPLOS ONE

Dear Dr. Vural,

Thank you for submitting your manuscript to PLOS ONE. After careful consideration, we feel that it has merit but does not fully meet PLOS ONE’s publication criteria as it currently stands. Therefore, we invite you to submit a revised version of the manuscript that addresses the points raised during the review process.

We look forward to receiving your revised manuscript.

Kind regards,

Raffaele Vitiello

Academic Editor

PLOS ONE

3. We note that there is identifying data in the Supporting Information file < S1 file_dataset.xlsx>. Due to the inclusion of these potentially identifying data, we have removed this file from your file inventory. Prior to sharing human research participant data, authors should consult with an ethics committee to ensure data are shared in accordance with participant consent and all applicable local laws.

-Location data

Reviewers' comments:

Reviewer's Responses to Questions

**Comments to the Author**

1. Is the manuscript technically sound, and do the data support the conclusions?

Reviewer #1: Yes

Reviewer #2: Yes

Reviewer #3: Yes

2. Has the statistical analysis been performed appropriately and rigorously? 

Reviewer #1: Yes

Reviewer #2: Yes

Reviewer #3: Yes

3. Have the authors made all data underlying the findings in their manuscript fully available?

Reviewer #1: Yes

Reviewer #2: Yes

Reviewer #3: Yes

4. Is the manuscript presented in an intelligible fashion and written in standard English?

Reviewer #1: Yes

Reviewer #2: Yes

Reviewer #3: Yes

5. Review Comments to the Author

Reviewer #1: The research question and methodology sound good. I have a few comments.

1) From line 124-126, could you explain more why you used cutoff points of 17.975 for HALP, 28.145 for CRP, and 0.6574 for CAR to predict the mortality?

2) Could you please provide more details about “Comorbidity”

3) Could you please provide more details about proximal femoral fracture: how many femoral neck and intertrochanteric fracture in this cohort?

4) The pictures were not clear. Could you please improve the quality of the pictures.

Reviewer #2: Authors have targeted a very sound and important subject which should be commended. Overall, the manuscript is well-written and analysis is very good. There are some suggestions that might increase overall comprehension.

1. Please add full abbreviation of HALP in the title.

2. Line no. 81-83: Please clarify this line: "which occur mostly in the setting of osteoporosis, and the HALP score", what do you mean by occur mostly in the HALP score. Elaboration will be appreciated.

3. Line no.134: Write statistical analysis as the heading.

4. Line no.181: correct spelling: "predicting"

5. Line no.208: Make separate table for univariate binary logistic regression analysis

6. Line no.232-235: Mention reference for the study.

7. Line no.235-238: Mention reference for cancer studies

Reviewer #3: The study titled, “HALP score for predicting early and late mortality in elderly patients with proximal femur fractures” by Vural et al. presents some very interesting findings on the HALP score and how it can be used to predict mortality. However, to further improve the manuscript I would suggest the authors make the following edits:

• Pages 2-3, Introduction Section: The introduction requires further improvement to clearly explain the need for a scoring system like HALP. The authors should highlight the limitations of existing scoring systems for predicting mortality in elderly patients with PFFs and explain how the HALP score addresses these gaps. Emphasizing the potential benefits and advantages of the HALP score would strengthen the rationale for its use in this context. Currently, the rationale for the study is weak and needs to be more compelling to justify the focus on the HALP score.

• Pages 2-3, Introduction Section: While one of the objectives of this study is to compare CRP and CAR with the HALP score in predicting mortality endpoints, the rationale behind selecting these specific markers for comparison is unclear. I suggest the authors elaborate on this to strengthen the relevance and significance of their study.

• Pages 4-5, Methods Section: Did the authors adjust the results for any confounding variables? If so, please mention them in the methodology section. If not, it should be noted in the limitations section of the study.

• Pages 8-9, Discussion Section: The authors have provided a thorough discussion comparing the HALP score with other predictive biomarkers, emphasizing its efficacy in predicting mortality in older patients with PFF. However, I believe that a deeper exploration of the biological mechanisms underlying the HALP score's effectiveness in predicting mortality would enhance the manuscript. Specifically, discussing how the components of the HALP score (hemoglobin, albumin, lymphocyte count, and platelet count) are linked to mortality could provide valuable insights into the clinical implications of the findings.

• Page 2, Line 50: Please recheck the entire manuscript for any typographical errors, such as the incorrect spelling of the abbreviation "PPF" in the abstract. Additionally, ensure that there is enough spacing between the sections of the manuscript.

• Page 10, Lines 104-105: The authors should explicitly mention what "Karmed" refers to in the methods section, as its current usage can lead to potential ambiguity in readers' minds.

6. PLOS authors have the option to publish the peer review history of their article (what does this mean?). If published, this will include your full peer review and any attached files.

Reviewer #1: No

Reviewer #2: **Yes: **Tabeer Tanwir Awan

Reviewer #3: No

---

## [Author Response · Author response to Decision Letter 0]

6 Oct 2024

To Academic Editor,

1. With our revision, we are confident that our article meets the style requirements of PLOS ONE, including file name.

2. Our ethical statement is included only in the Methods section of our article.

3. The Supporting Information file <S1 file_dataset.xlsx> has been fully anonymised according to your suggestions. For this purpose, simple identifying information that could potentially identify participants (such as gender) and sensitive information such as date of surgery, date of death have been removed from the data set. In this way, our data has been completely anonymised.

4. Our reference list has been reviewed again. We are confident that our References are complete and accurate, with new references added. There are no references to retracted articles.

First, we thank the reviewers for their valuable contributions.

Reviewer #1: The research question and methodology sound good. I have a few comments.

1) From line 124-126, could you explain more why you used cutoff points of 17.975 for HALP, 28.145 for CRP, and 0.6574 for CAR to predict the mortality?

Response: We used Receiver Operating Characteristic (ROC) curve analysis to determine the most appropriate thresholds for predicting early and late mortality. ROC curves allow us to evaluate the diagnostic performance of a biomarker by balancing sensitivity (true positive rate) and specificity (true negative rate) at various cut-off points. The Youden index was chosen as the most appropriate cut-off point because it maximises the sum of sensitivity and specificity and identifies the point that best discriminates between mortality and survival. By maximising this index, we ensured that the cut-off points we selected for HALP, CRP and CAR offered the best balance between sensitivity and specificity and optimised their predictive power for mortality. This method is particularly valuable in clinical settings where determining the most reliable threshold is critical for timely and accurate prognostic assessment. Thus, the thresholds of 17.975 for HALP, 28.145 for CRP and 0.6574 for CAR were not arbitrarily chosen but were based on the highest Youden index and provided the highest accuracy in survival analysis (Kaplan-Meier method, Log-rank test) in predicting mortality outcomes in the studied cohort. This explanation is included in detail in the relevant section of the article (Line 147-159).

2) Could you please provide more details about “Comorbidity”

Response: The most common chronic diseases in the study cohort were hypertension, diabetes, COPD, CAD and stroke, and patients were categorised as having or not having at least one of these diseases. Detailed data are given in the Results section and Table 1.

3) Could you please provide more details about proximal femoral fracture: how many femoral neck and intertrochanteric fracture in this cohort?

Response: 257 patients (59%) had femoral neck fractures and 178 patients (41%) had intertrochanteric fractures. It was added to the result section of the study.

4) The pictures were not clear. Could you please improve the quality of the pictures.

Response: The image quality of Figure 1 has been improved. 

Reviewer #2: Authors have targeted a very sound and important subject which should be commended. Overall, the manuscript is well-written and analysis is very good. There are some suggestions that might increase overall comprehension.

1. Please add full abbreviation of HALP in the title.

Response: The full name and abbreviation of HALP is given in the title.

2. Line no. 81-83: Please clarify this line: "which occur mostly in the setting of osteoporosis, and the HALP score", what do you mean by occur mostly in the HALP score. Elaboration will be appreciated.

 Response: This statement has been reorganised and the confusion removed.

3. Line no.134: Write statistical analysis as the heading.

 Response: Corrected.

4. Line no.181: correct spelling: "predicting"

 Response: Corrected.

5. Line no.208: Make separate table for univariate binary logistic regression analysis

Response: Added. ( Note: It was understood that the 1-year mortality risk ratios for the HALP score were given according to the quantitative HALP scores and comments were made in this direction, and the correct ratios were provided by re-categorising according to the determined threshold value (17.975). The inadvertent error has been corrected.)

6. Line no.232-235: Mention reference for the study.

Response: References (7, 32-37) mentioned for studies. Line 297-301.

7. Line no.235-238: Mention reference for cancer studies

Response: This statements are contextualised by combining it with the previous statements. no additional reference is needed.

Reviewer #3: The study titled, “HALP score for predicting early and late mortality in elderly patients with proximal femur fractures” by Vural et al. presents some very interesting findings on the HALP score and how it can be used to predict mortality. However, to further improve the manuscript I would suggest the authors make the following edits:

• Pages 2-3, Introduction Section: The introduction requires further improvement to clearly explain the need for a scoring system like HALP. The authors should highlight the limitations of existing scoring systems for predicting mortality in elderly patients with PFFs and explain how the HALP score addresses these gaps. Emphasizing the potential benefits and advantages of the HALP score would strengthen the rationale for its use in this context. Currently, the rationale for the study is weak and needs to be more compelling to justify the focus on the HALP score.

Response: The introduction has been expanded to clearly explain the need for the HALP score. The limitations of current scoring systems for predicting mortality in elderly patients with PFF are highlighted and it is explained how the HALP score addresses these shortcomings. The potential benefits and advantages of the HALP score are emphasised. In this regard, references 17 and 18 have been added.

• Pages 2-3, Introduction Section: While one of the objectives of this study is to compare CRP and CAR with the HALP score in predicting mortality endpoints, the rationale behind selecting these specific markers for comparison is unclear. I suggest the authors elaborate on this to strengthen the relevance and significance of their study.

Response: In this study, the relevance and importance of the study were strengthened by justifying the rationale behind the selection of these specific markers to compare the HALP score with CRP and CAR. In this study, the ability of CRP, CAR and HALP scores to predict mortality outcomes was analysed by comparing these parameters. The rationale for this is that each biomarker provides different clinical information and their combined evaluation may provide a more comprehensive prognosis. The HALP score, which includes haemoglobin, albumin, lymphocyte count and platelet count, provides information on inflammation and nutritional status. However, as CRP and CAR can directly reflect levels of systemic inflammation, their comparison with the HALP score may provide a more holistic approach to mortality risk, particularly in elderly patients, and may help to develop more effective management strategies for elderly patients with hip fracture.

• Pages 4-5, Methods Section: Did the authors adjust the results for any confounding variables? If so, please mention them in the methodology section. If not, it should be noted in the limitations section of the study.

Response: Any confounding variables that may have an effect on the results of the study were not addressed under subgroups. This is explained in detail in the limitations section of the study.

• Pages 8-9, Discussion Section: The authors have provided a thorough discussion comparing the HALP score with other predictive biomarkers, emphasizing its efficacy in predicting mortality in older patients with PFF. However, I believe that a deeper exploration of the biological mechanisms underlying the HALP score's effectiveness in predicting mortality would enhance the manuscript. Specifically, discussing how the components of the HALP score (hemoglobin, albumin, lymphocyte count, and platelet count) are linked to mortality could provide valuable insights into the clinical implications of the findings.

Response: We discussed how the components of the HALP score (hemoglobin, albumin, lymphocyte count and platelet count) are associated with mortality. The discussion was improved by adding new references. In this way, the importance of the HALP score was further strengthened.

• Page 2, Line 50: Please recheck the entire manuscript for any typographical errors, such as the incorrect spelling of the abbreviation "PPF" in the abstract. Additionally, ensure that there is enough spacing between the sections of the manuscript.

Response: The article was re-evaluated in terms of spelling and grammar rules. Spelling errors (including abbreviations) were corrected.

• Page 10, Lines 104-105: The authors should explicitly mention what "Karmed" refers to in the methods section, as its current usage can lead to potential ambiguity in readers' minds.

Response: In order to avoid confusion, the word ‘karmed’ was removed and it was clearly written what ‘karmed’ is.

---

## [Decision Letter · Decision Letter 1]

1 Nov 2024

Hemoglobin, albumin, lymphocyte and platelet (HALP) score for predicting early and late mortality in elderly patients with proximal femur fractures

PONE-D-24-38419R1

Dear Dr. Vural,

We’re pleased to inform you that your manuscript has been judged scientifically suitable for publication and will be formally accepted for publication once it meets all outstanding technical requirements.

Kind regards,

Raffaele Vitiello

Academic Editor

PLOS ONE

Additional Editor Comments (optional):

Reviewers' comments:

Reviewer's Responses to Questions

**Comments to the Author**

1. If the authors have adequately addressed your comments raised in a previous round of review and you feel that this manuscript is now acceptable for publication, you may indicate that here to bypass the “Comments to the Author” section, enter your conflict of interest statement in the “Confidential to Editor” section, and submit your "Accept" recommendation.

Reviewer #2: (No Response)

Reviewer #3: All comments have been addressed

2. Is the manuscript technically sound, and do the data support the conclusions?

Reviewer #2: (No Response)

Reviewer #3: Yes

3. Has the statistical analysis been performed appropriately and rigorously? 

Reviewer #2: (No Response)

Reviewer #3: Yes

4. Have the authors made all data underlying the findings in their manuscript fully available?

Reviewer #2: (No Response)

Reviewer #3: Yes

5. Is the manuscript presented in an intelligible fashion and written in standard English?

Reviewer #2: (No Response)

Reviewer #3: Yes

6. Review Comments to the Author

Reviewer #2: (No Response)

Reviewer #3: (No Response)

7. PLOS authors have the option to publish the peer review history of their article (what does this mean?). If published, this will include your full peer review and any attached files.

Reviewer #2: **Yes: **Tabeer Tanwir Awan

Reviewer #3: No

---

## [Editor Report · Acceptance letter]

11 Nov 2024

PONE-D-24-38419R1 

PLOS ONE

Dear Dr. Vural, 

I'm pleased to inform you that your manuscript has been deemed suitable for publication in PLOS ONE. Congratulations! Your manuscript is now being handed over to our production team.

Kind regards, 

on behalf of

Dr. Raffaele Vitiello 

Academic Editor

PLOS ONE